# Failure to modulate reward prediction errors in declarative learning with theta (6 Hz) frequency transcranial alternating current stimulation

**Kate Ergo**[1]*, **Esther De Loof**[1], **Gillian Debra**[1], **Bernhard Pastötter**[2], **Tom Verguts**[1]

**1** Department of Experimental Psychology, Ghent University, Ghent, Belgium, **2** Department of Psychology, University of Trier, Trier, Germany

* kate.ergo@ugent.be

**Data Availability Statement:** The data are publicly available on OSF (DOI 10.17605/OSF.IO/ZXHQ4).

**Funding:** KE conducted the research as a doctoral researcher, supported by grant 1153418N of the

## Abstract

Recent evidence suggests that reward prediction errors (RPEs) play an important role in declarative learning, but its neurophysiological mechanism remains unclear. Here, we tested the hypothesis that RPEs modulate declarative learning via theta-frequency oscillations, which have been related to memory encoding in prior work. For that purpose, we examined the interaction between RPE and transcranial Alternating Current Stimulation (tACS) in declarative learning. Using a between-subject (real versus sham stimulation group), single-blind stimulation design, 76 participants learned 60 Dutch-Swahili word pairs, while theta-frequency (6 Hz) tACS was administered over the medial frontal cortex (MFC). Previous studies have implicated MFC in memory encoding. We replicated our previous finding of signed RPEs (SRPEs) boosting declarative learning; with larger and more positive RPEs enhancing memory performance. However, tACS failed to modulate the SRPE effect in declarative learning and did not affect memory performance. Bayesian statistics supported evidence for an absence of effect. Our study confirms a role of RPE in declarative learning, but also calls for standardized procedures in transcranial electrical stimulation.

## Introduction

Declarative memory consists of memory for facts and events that can be consciously recalled [1, 2]. Memoranda are learned rapidly, often after a single exposure [3]. The process of acquiring such memories is called declarative learning. Declarative memory differs from procedural memory, where a skill is learned slowly and by means of repeated practice (e.g., learning how to drive a car). Research has firmly established that prediction errors modulate declarative memory [4], just like they do in procedural memory [5]. Recent research shows that reward prediction errors (RPE; i.e., mismatches between reward outcome and reward prediction) specifically may facilitate memory formation. RPEs were primarily studied within procedural learning (e.g., [6]). However, recent evidence suggests that RPEs are crucial for declarative learning as well [7–9].

Research Foundation Flanders (https://www.fwo.
be/). EDL and TV were supported by grant BOF17-
GOA-004 from the Research Council of Ghent
University (https://www.ugent.be/nl/onderzoek/
financiering/bof). The funders had no role in study
design, data collection and analysis, decision to
publish, or preparation of the manuscript.

**Competing interests:** The authors have declared
that no competing interests exist.

One robust experimental paradigm to test this RPE effect on declarative memory, was proposed in [10]. Here, a variable-choice experimental paradigm was used where participants learned Dutch-Swahili word pairs. On each trial, participants were presented with one Dutch word and four Swahili translations. By fixing a priori the number of eligible Swahili translations and whether a choice was rewarded or not, each trial was associated with a different RPE. As a consequence, participants did not learn the actual Swahili translations for the Dutch words. This manipulation allowed verifying whether declarative learning was driven by unsigned RPEs (URPE; signifying that the outcome is different than expected) or instead by signed RPEs (SRPE; indicating that the outcome is better or worse than expected). If URPEs boost declarative learning, recognition of word pairs should be enhanced for large positive and large negative RPE values, exhibiting a U-shaped effect of RPE on memory. Instead, if SRPEs drive declarative learning, recognition should be increased only for large, positive RPEs. The data revealed a SRPE effect. Larger and more positive RPEs during study improved subsequent declarative memory during testing. The effect of RPEs in this experimental paradigm was further substantiated in a follow-up EEG study, where oscillatory signatures at reward feedback were detected in the theta (4–8 Hz), high-beta (20–30 Hz) and high-alpha (10–15 Hz) frequency ranges, suggesting the experience of RPEs by the participants [11]. Further validation came from an fMRI study using a similar paradigm in which famous faces were associated with Swahili village names [12]. This study revealed that RPE responses in the ventral striatum (VS) at reward feedback predicted memory performance. These findings lend further support to the notion that RPE is a key factor in the formation of new declarative memories, and that RPEs are characterized by distinctive neural signatures.

It remains unclear, however, how RPEs boost declarative memory. It is well established that RPEs are encoded by dopaminergic neurons in the midbrain (i.e., ventral tegmental area and substantia nigra) [5]. These neurons change their firing rate in relation to RPEs. From the midbrain, RPEs are projected to several other subcortical and cortical brain regions, such as the VS [13], the hippocampus (HC) [14], and the medial frontal cortex (MFC) [15]. Within these brain structures, dopamine release functions as a neuromodulatory signal. One potential neuromodulatory influence of dopamine occurs via modulating neural oscillations in a wide range of frequency bands [16]. Neural activity in the theta frequency band (4–8 Hz) seems to be of particular importance in memory encoding [17]. Indeed, oscillations in the theta frequency allow communication between distant brain regions, promote encoding of novel information [18], enable learning [19], and have been linked to improved declarative memory [20–22].

One possible mechanism through which theta frequency improves memory is theta phase synchronization. Synchronization in declarative memory can be observed locally, for example, using intracranial electrodes placed in the medial temporal lobe. With this method [23], found increased theta phase locking during the encoding of words. Theta phase synchronization can also be observed non-locally. When multimodal (audio-visual) stimuli are synchronously presented in theta phase, episodic memory is enhanced; with stronger theta phase synchronization between the visual and auditory cortex predicting better memory performance [24, 25]. Furthermore [26], observed increased theta phase synchronization between HC and prefrontal cortex (PFC) during the presentation of unexpected items. Interestingly, the PFC, and in particular the MFC, has been ascribed an important role in memory encoding [27–29]. It is also strongly implicated in reward [30, 31] and RPE [32, 33] processing. We hypothesize that during declarative learning, RPEs project to the MFC [15], where they are used to optimize future behavior [34]. Specifically, RPEs may (by means of neuromodulatory signaling) increase theta (phase) synchronization between relevant brain areas (e.g., MFC and HC), therefore allowing associative memories to be glued together more efficiently [35], facilitating (multimodal) memory formation [36].

Unfortunately, the evidence for theta modulation of RPEs in declarative memory thus far remains correlational only. With the rise of non-invasive brain stimulation (NIBS) techniques, the causal role of neural oscillations and their relation to behavior can be explicitly tested [37]. More specifically, transcranial Alternating Current Stimulation (tACS) allows modulating neural oscillations [38]. It is hypothesized that tACS causes underlying brain networks to synchronize or desynchronize. Although tACS has rather low temporal and spatial resolution, its frequency resolution is high. By applying a weak sinusoidal current to the scalp, the likelihood of neural firing is increased or decreased, depending on the stimulation parameters [39]. Ongoing neural oscillations can thus be entrained at specific frequencies of interest [39]. This synchronization modulates brain activity and alters cognitive processes, leading to behavioral changes, which can be measured through, for example, memory performance [40].

Whereas several tACS experiments entraining oscillations at theta frequency looked at its effects on working memory [41–46], a few studies have investigated its effects on declarative memory [47]. applied theta-frequency tACS over the right fusiform cortex while face and scene pairs were encoded. Here, stimulation enhanced memory performance measured after a 24-hour delay. Similarly [48], also found enhanced long-term memory performance after applying theta-frequency tACS over the right posterior cortex while participants learned face-monetary value pairs. To the best of our knowledge, no study examined the effects of theta-frequency tACS over MFC in relation to declarative learning.

Together, these findings suggest that RPEs are projected from brainstem to MFC; elicit theta phase synchronization between several neural areas; and thus boost declarative learning. As such, the goal of the current study was to use theta-frequency (6 Hz) tACS to entrain neural oscillations whilst encoding new word pairs associated with RPEs of different sizes and values. To this end, tACS was applied over the MFC while participants acquired 60 Dutch-Swahili word pairs using the variable-choice experimental paradigm. We hypothesized that if declarative learning is modulated by theta oscillations in MFC, then subsequent memory performance and certainty ratings should be modulated by tACS (i.e., higher recognition accuracies and certainty ratings in the real compared to sham stimulation group); and if theta oscillations are driven by RPE, as the literature review suggests, tACS and RPE should interact.

## Methods

### Participants

We tested a total of 77 healthy, Dutch-speaking participants. One participant was excluded from further analysis due to below chance level performance on the recognition test. The analyses were run on the remaining 76 participants (57 females, range = 18–29 years, $M_{age}$ = 20.8 years, $SD_{age}$ = 2.4 years). All participants had no prior knowledge of Swahili, gave written informed consent, were randomly assigned to a real (N = 38) or sham (N = 38) stimulation group, and were paid €17.5. The study was approved by the Medical Ethics Review Board of the Ghent University Hospital and was carried out in accordance with the Declaration of Helsinki.

### Material

A total of 330 words (66 Dutch, 24 Japanese and 240 Swahili words) (S1–S4 Tables) were used. Each participant memorized 60 Dutch-Swahili word pairs. The experiment was run on an HP ProBook 6560b laptop with a 15.6" screen size running PsychoPy software (version 1.85.4) [49].

## Experimental paradigm

**Familiarization task.** Participants started with a familiarization task using the stimuli in the experiment, to control for the novelty of the foreign Swahili words. All Dutch (N = 60) and Swahili (N = 240) words were randomly and sequentially presented on the screen for a duration of two seconds. Participants were asked to press the space bar whenever a Dutch word was presented.

**Acquisition task.** Prior to the actual acquisition task, a total of six practice trials with Dutch (N = 6) and Japanese (N = 24) words was presented. After successfully finishing the practice set, participants were presented with the acquisition task. Here, the aim was to learn 60 unique Dutch-Swahili word pair associations. On each trial, one Dutch word was shown together with four Swahili translations (Fig 1A). After four seconds, frames surrounded the eligible Swahili translations. Either one, two or four Swahili translations were framed. In the one-option condition, one Swahili translation was framed and participants could only choose this Swahili word as the translation for the Dutch word. In the two-option condition, two Swahili translations were framed and participants could choose between two options. In the four-option condition trials, all four Swahili translations were framed and participants could choose among these four options. The probability of choosing the correct Swahili translation was therefore 100% (in one-option condition trials), 50% (in two-option condition trials), or 25% (in four-option condition trials). Importantly, each trial was associated with a specific RPE value by fixing a priori whether a trial was rewarded or not and the number of eligible Swahili translations. As a result, participants did not learn the actual Swahili translations of the Dutch words. They were unaware of this manipulation during the experiment, but were debriefed afterwards. Note also that although not explicitly communicated to the participants, there was a clear, normatively correct choice that had to be remembered on each trial. The intention of the experiment was also made clear by the colors (i.e., red/green) and the feedback (i.e., wrong/correct) that were used in the acquisition task. Participants responded with the index and middle finger of the right and left hand. For stimulation purposes, trial duration was controlled by instructing participants to make their choice as soon as the fixation cross turned blue. If no choice was made after two seconds, the fixation cross turned red, urging participants to choose as soon as possible. To ensure that stimulation was given throughout the entire duration of the acquisition task, total time spent in the acquisition task was equated for each participant. Specifically, if participants made a choice less than two seconds after the fixation cross turned blue, feedback was presented after [two seconds—choice duration] seconds. After participants made their choice, the fixation cross turned into a blue "o" indicating that their response had been registered. They were then provided with feedback where they saw the Dutch word, an equation sign, and the to-be-learned Swahili translation (in green for correct choices and in red for incorrect choices) for a duration of five seconds. This was followed by reward feedback (+0.5 Euros for correct choices and +0 Euros for incorrect choices) and a reward update telling them how much money they earned up until the last completed trial (two seconds). After every ten trials, the acquisition task was briefly paused for ten seconds to allow an impedance check.

*Design*. Parametric modulation of RPEs was accomplished by fixing a priori the number of options (one, two or four) and reward on each trial (reward/no reward). This allowed the computation of an RPE for each cell of the design (Fig 1B). In addition, the proportion of trials in each cell of the design matched the reward expectation (i.e., 100% rewarded trials in the one-option condition, 50% rewarded and 50% non-rewarded trials in the two-option condition, and 25% rewarded and 75% non-rewarded trials in the four-option condition).

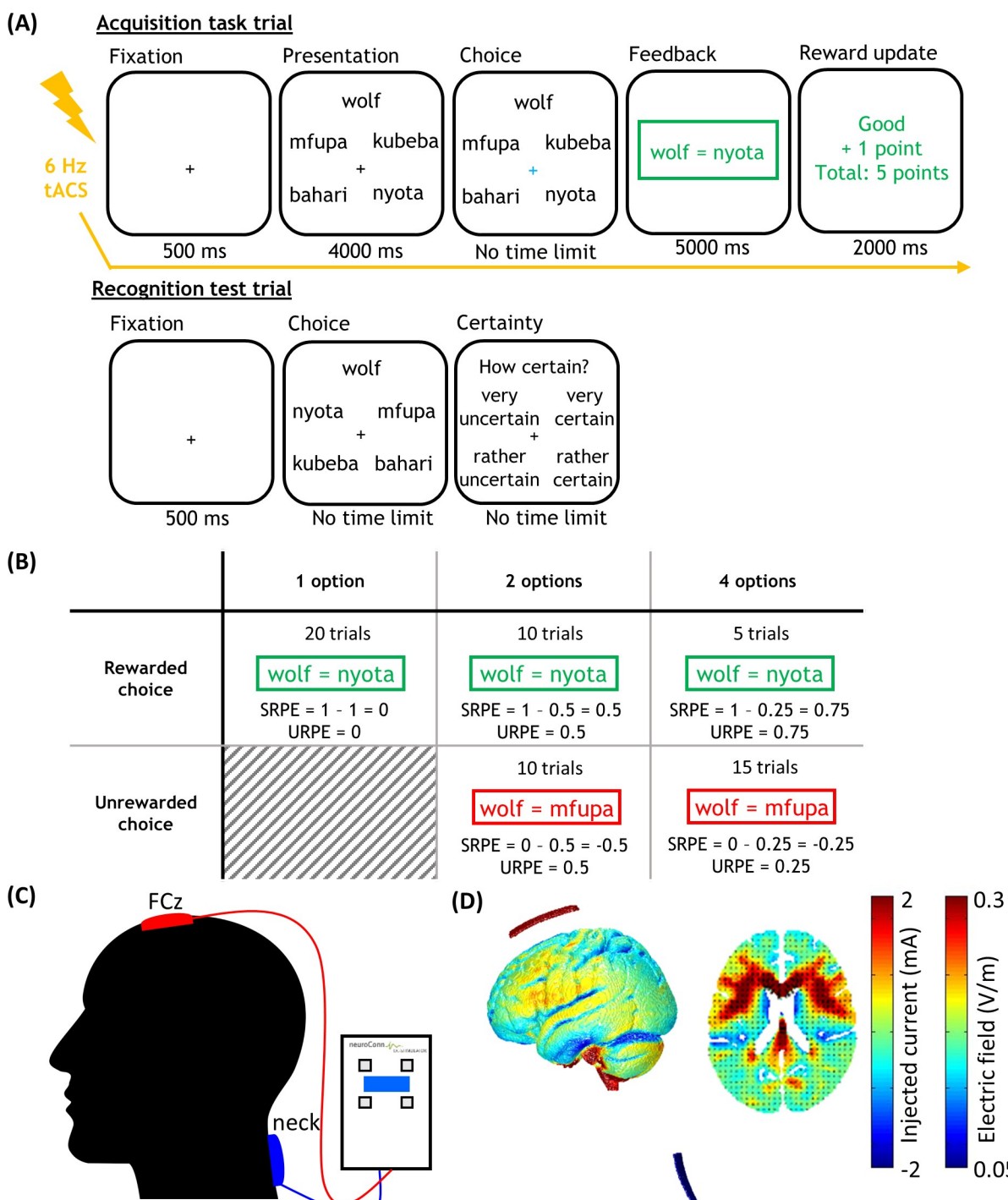

**Fig 1. Experimental paradigm and tACS setup.** (A) Example trial of the acquisition task and recognition test. In the acquisition task, participants choose between 1, 2 or 4 Swahili translations. The two-option condition with rewarded choice is illustrated. (B) Experimental design. The 2 (rewarded or unrewarded choice) x 3 (number of options) experimental design showing the number of trials and associated RPE value in each cell. SRPEs were calculated by subtracting the probability of reward from the obtained reward; URPE is the absolute value of SRPE. (C) tACS setup. Theta-frequency (6 Hz) tACS was applied over the MFC. The stimulation electrode (i.e., red electrode) was placed over FCz, while the reference electrode (i.e., blue electrode) was placed on the neck. (D) Simulation of the electric field with the ROAST toolbox.

SRPEs were obtained by subtracting reward probability from reward outcome. For rewarded trials, reward outcome is equal to one, whereas reward outcome is equal to zero for unrewarded trials. Reward probability is determined by the number of options. URPEs are computed by taking the absolute value of the SRPE.

**Recognition test.** In the recognition test, participants' recognition was tested on 60 Dutch-Swahili word pairs that were acquired during the acquisition task (Fig 1A). On each trial, one Dutch word was shown together with the same four Swahili translations from the acquisition task. Spatial positions of the Swahili translations were randomly shuffled relative to the acquisition task to avoid that participants would respond based on the spatial position instead of the learned translation of the Dutch word. In contrast to the acquisition task, no frames surrounded the Swahili translations, and no feedback was provided. No time limit was imposed. At the end of each trial, participants rated their certainty on a four-point scale ("very certain", "rather certain", "rather uncertain", "very uncertain").

## Sensations questionnaire

A subset of participants (N = 61) filled out a sensations questionnaire [50] (S1 File). Participants rated seven sensations (itching, pain, burning, warmth/heat, pinching, metallic/iron taste and fatigue) on a five-point scale (none, mild, moderate, considerable, strong). They were also asked when the discomfort began, how long the discomfort lasted and how much these sensations affected their performance. The sensations questionnaire was used to verify whether participants in the real and sham stimulation group report a difference in sensations.

## tACS stimulation

tACS stimulation was applied using a DC-stimulator Plus device (NeuroConn GmbH, Ilmenau, Germany). Two saline-soaked sponge electrodes (5 x 6.5 cm$^2$) were placed on the scalp and neck. The stimulation (red) electrode was positioned at FCz (according to the 10–20 positioning system), targeting the MFC, while the reference (blue) electrode was placed on the neck (Fig 1C). The sponge electrodes were fixed onto the participant's head with elastic fabric bands. Impedance between electrodes was kept below 15 kΩ. Participants received tACS stimulation at the theta (6 Hz) frequency with an intensity of 2 mA (peak-to-peak; mean 0 mA). A sinusoidal stimulation waveform was used with no DC offset and a phase shift of zero degrees. A fade-in and fade-out period of 5 seconds (30 cycles) was used. tACS was administered during the entire acquisition task for a duration of 16.6 minutes (6000 cycles) in the real stimulation group, while the sham stimulation group received 40 seconds (240 cycles) of stimulation at the beginning of the acquisition task only. Sham stimulation duration was deliberately kept short to avoid changes in cortical excitability [51, 52]. Current flow was simulated using the ROAST (Realistic vOlumetric Approach to Simulate Transcranial electric stimulation) toolbox [53] in MATLAB (Fig 1D).

## Data analysis

Both frequentist and Bayesian statistics were calculated. With regard to frequentist statistics, all data were analyzed within the linear mixed effects framework in R software [54], unless mentioned otherwise. For continuous dependent variables (i.e., certainty ratings in the recognition test) linear mixed effects models were used, while for categorical dependent variables (i.e., recognition accuracy) generalized linear mixed effects models were applied. A random intercept for participants was included in each model, while all predictors (i.e., accuracy, SRPE and stimulation) were mean-centered. Note that SRPEs were treated as a continuous predictor allowing the inclusion of all 60 trials per participant to estimate its regression coefficient, with

the exception of invalid trials (i.e., trials on which a non-framed Swahili translation was chosen during the acquisition task). We report the $\chi^2$ statistics from the ANOVA Type III tests. All data are made publicly available at OSF (DOI 10.17605/OSF.IO/ZXHQ4).

In addition to frequentist statistics, Bayesian repeated measures analyses of variance (ANOVAs) are reported that were performed in JASP (version 0.11.1; [55]). In Bayesian ANOVAs, recognition accuracy and certainty ratings were analyzed as a function of SRPE and stimulation. Bayes factors (BFs) quantify the evidence in favor of the null hypothesis ($BF_{01}$; e.g., tACS does not influence memory performance) or the alternative hypothesis ($BF_{10} = 1/BF_{01}$; e.g., tACS influences memory performance). $BF_{01}$ is reported when the Bayesian analysis provides relatively more evidence for the null hypothesis; $BF_{10}$ is instead reported when the analysis provides relatively more evidence for the alternative hypothesis. We used default prior settings for all analyses [56]. To determine the strength of evidence, we used Jeffreys' benchmarks [57], with BFs corresponding to anecdotal (0–3), substantial (3–10), strong (10–30), very strong (30–100) or decisive (>100) evidence.

## Results

### Sensations questionnaire

Independent samples t-tests were used to verify whether sensations varied between the two stimulation groups. Participants in the real and sham stimulation groups did not report a significant difference for any of the sensations probed (itching, pain, burning, warmth/heat, pinching, metallic/iron taste and fatigue) (all $p > .06$). Furthermore, there were no significant differences between stimulation groups with regard to when the discomfort began, $t(58.90) = 0.48$, $p = .63$ (real: $M = 1.23$, $SD = 0.50$, $range = 0–2$; sham: $M = 1.17$, $SD = 0.46$, $range = 0–2$), and how much these sensations affected their performance, $t(53.77) = 1.13$, $p = .26$ (real: $M = 1.39$, $SD = 0.62$, $range = 0–4$; sham: $M = 1.23$, $SD = 0.43$, $range = 0–4$). Participants in the real stimulation group did report that the discomfort lasted significantly longer compared to the sham stimulation group, $t(40.33) = 3.35$, $p = .002$ (real: $M = 1.68$, $SD = 0.83$, $range = 0–2$; sham: $M = 1.13$, $SD = 0.35$, $range = 0–2$).

### Recognition accuracy

Here, we verified whether recognition accuracy linearly increased with SRPEs. Replicating earlier research, frequentist statistics revealed a significant positive effect of SRPE, $\chi^2(1, N = 76) = 9.13$, $p = .003$, with larger and more positive RPEs leading to increased recognition accuracy (Fig 2A and 2B). There was no main effect of stimulation on recognition accuracy, $\chi^2(1, N = 76) = 1.42$, $p = .23$. The interaction between SRPE and stimulation was also not significant, $\chi^2(1, N = 76) = .004$, $p = 0.95$.

Bayesian repeated measures ANOVA provided substantial evidence for the absence of a stimulation effect ($BF_{01} = 3.02$, evidence for null versus alternative model). Thus, the observed data were about 3 times more likely under the model that included no stimulation than under the alternative model that did. The evidence for the SRPE effect was decisive ($BF_{10} > 100$, evidence for alternative versus null model). In addition, there was strong evidence against the interaction of SRPE and stimulation ($BF_{01} = 54.66$, evidence for main-effects-only relative to main-effects-plus-interaction model).

### Certainty ratings

For the certainty ratings there was a significant main effect of recognition accuracy, $\chi^2(1, N = 76) = 1170$, $p < .001$, indicating that participants were more certain of correctly recognized

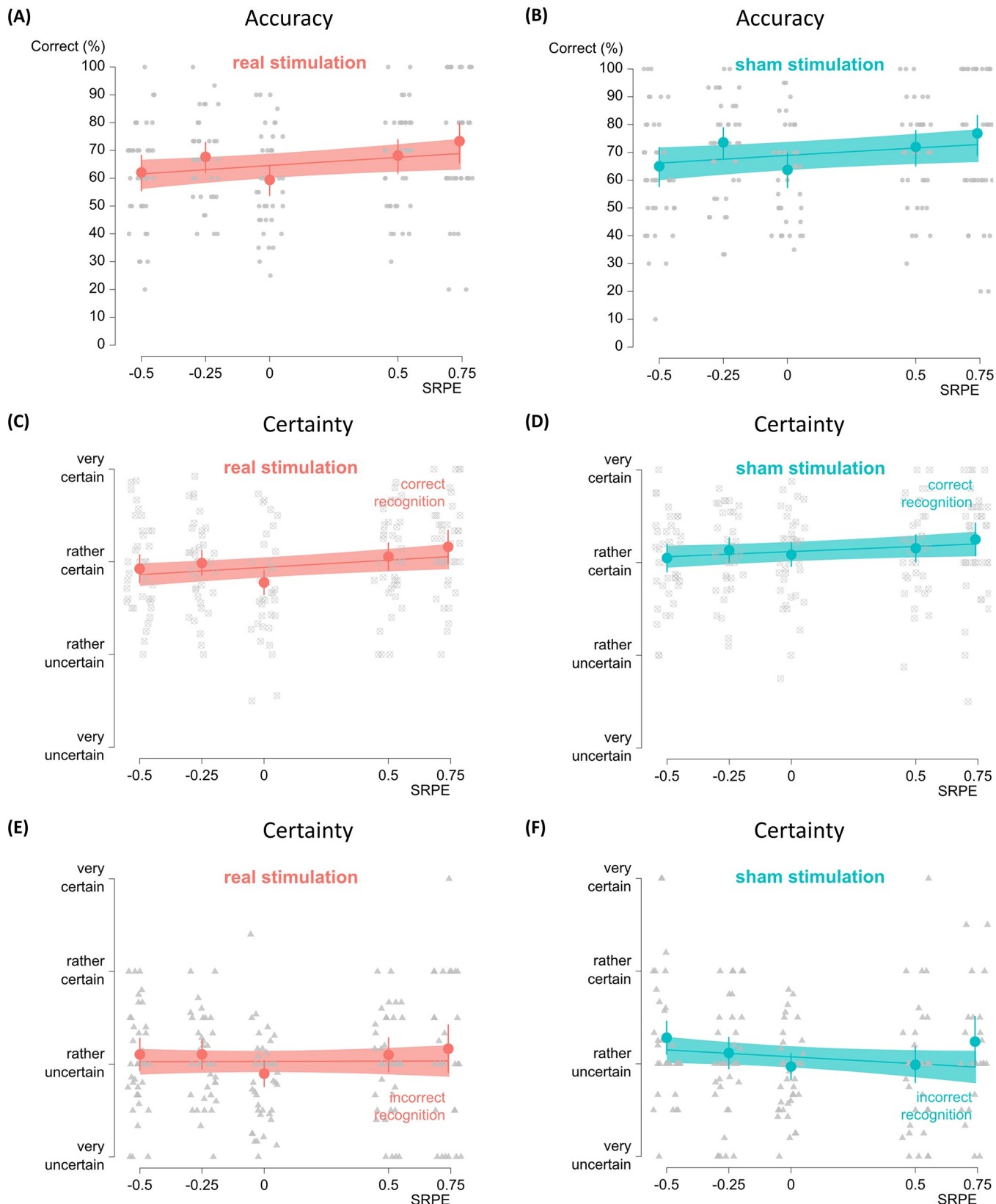

**Fig 2. Results.** (A-B) Recognition accuracy as a function of SRPE in the real and sham stimulation group, respectively. The average recognition and its 95% confidence interval were estimated and superimposed. Gray dots represent data points for individual subjects. Recognition accuracy increases linearly with larger and more positive RPEs in the two stimulation groups, suggesting a SRPE effect. (C-D) Certainty rating for correct recognitions in the real and sham stimulation group, respectively. The average certainty and its 95% confidence interval were estimated and superimposed. Gray dots and rectangles represent data of individual subjects for correct recognitions. In the two stimulation groups, SRPE significantly predicted certainty for correctly recognized word pairs. (E-F) Certainty rating for incorrect recognitions in the real and sham stimulation group, respectively. The average certainty and its 95% confidence interval were estimated and superimposed. Gray dots and rectangles represent data of individual subjects for incorrect recognitions. In the two stimulation groups, SRPE did not significantly predict certainty for incorrectly recognized word pairs.

word pairs compared to incorrectly recognized word pairs (see (S1–S4 Figs) for within-subject behavioral responses for the certainty ratings). In addition, there was a significant interaction between SRPE and recognition accuracy, $\chi^2(1, N = 76) = 7.63$, $p = .006$. Follow-up analysis revealed that, as expected, SRPE increased certainty for correctly recognized word pairs, $\chi^2(1, N = 76) = 9.14$, $p = .002$, but did not affect certainty for false recognitions, i.e., incorrectly recognized word pairs, $\chi^2(1, N = 76) = 2.16$, $p = .14$ (Fig 2C and 2D). In addition, the data revealed a significant interaction between stimulation and recognition accuracy on certainty ratings, $\chi^2(1, N = 76) = 5.37$, $p = .02$. Follow-up analysis revealed a main effect of stimulation for the correctly recognized word pairs, $\chi^2(1, N = 76) = 5.03$, $p = .02$, but not for incorrectly recognized word pairs, $\chi^2(1, N = 76) = 0.11$, $p = .75$. Participants in the sham stimulation group were more certain of correctly recognized word pairs, compared to participants in the real stimulation group. Importantly, although participants in the real stimulation group reported increased discomfort duration, the effect of discomfort duration did not significantly affect certainty in the real, $\chi^2(1, N = 31) = 0.93$, $p = .33$, and sham, $\chi^2(1, N = 30) = 0.19$, $p = .66$, stimulation groups. This suggests that discomfort in itself did not influence the certainty rating. Finally, the interaction between SRPE and stimulation was not significant, $\chi^2(1, N = 76) = 1.61$, $p = .20$.

A Bayesian repeated measures ANOVA revealed anecdotal evidence for the absence of a stimulation effect ($BF_{01} = 1.33$, null model relative to model including stimulation). For the SRPE effect, the evidence was decisive ($BF_{10} > 100$, model including SRPE compared to null model). We also found strong evidence against the interaction of SRPE and stimulation ($BF_{01} = 19.74$, compared to two-main-effects model).

## Discussion

The main objective of our study was to examine if theta-frequency (6 Hz) tACS can modulate the effect of RPEs in declarative learning. For this purpose, participants acquired 60 Dutch-Swahili word pairs, associated with RPEs of different sizes and values, while the MFC was stimulated. We replicated our earlier finding of SRPEs driving declarative learning [10]. Word pair recognition increased for large and positive RPEs. However, contrary to our hypothesis, theta-frequency (6 Hz) tACS did not successfully improve memory nor modulate the effect of RPEs on declarative learning. There was a small effect of stimulation on certainty in the correctly recognized words, but this effect requires replication and must currently be interpreted with caution.

Whereas the importance of RPEs in procedural learning has been well established, its role in declarative learning has remained elusive until recently. One of the first experimental paradigms examining the effect of RPEs in declarative learning was put forward by [58]. Although this RPE effect on declarative memory could not be replicated [59, 60], several research labs have since then used a range of experimental paradigms to investigate the role of RPEs in declarative learning. Most of these studies revealed positive effects of RPEs on declarative memory [8, 9, 61], but one study also reported negative effects [62] (for review see [7]).

Overall, these studies (including the current one) support the claim that RPEs are a key factor in the formation of declarative memory.

Prior research has repeatedly shown a role of theta frequency in (reward) prediction error processing [63–66] as well as memory performance [21]. In particular [25], provided direct evidence for a causal role of theta frequency in memory. Memory for multimodal (audio-visual) stimuli was enhanced only when these stimuli were modulated at the theta frequency and not at other frequencies. Furthermore, in an earlier EEG study from our lab, we examined the neural signatures of RPEs in declarative learning and found increased theta (4–8 Hz) power during reward feedback [11]. However, it must be noted that in this particular EEG study, theta frequency followed an unsigned RPE (URPE) pattern during reward feedback. Theta power thus increased for both large negative and large positive RPEs. This URPE pattern evolved into a SRPE pattern during reward feedback and was accompanied by power increases in the high-beta (20–30 Hz) and high-alpha (10–17 Hz) frequency bands. Although beta and alpha power followed a clear SRPE pattern, we opted not to stimulate at these frequencies as there is more inter-individual variability with regard to peak-frequency [67].

We hypothesized that declarative learning is facilitated by theta frequency synchronization. Neurons are synchronized when their activation is locked to a common (slow-wave) phase. In such case, spikes of pre- and postsynaptic neurons are highly correlated, enabling synaptic learning between pairs of neurons because synaptic plasticity relies on the precise spike-timing of neurons [68]. Theta phase may modulate spike-timing-dependent plasticity by ensuring that (anatomically distant) neurons fire in synchrony [69, 70]. As tACS modulates the spike-timing of neurons [71–73], it is a promising tool to causally manipulate neural oscillations related to RPE-processing in declarative learning. For this reason, theta-frequency tACS was used to stimulate the MFC. Unfortunately, however, our tACS manipulation did not affect memory performance.

In the following section, we speculate why we found no effect of theta-frequency (6Hz) tACS and provide suggestions for future research. First, tACS has a relatively low spatial resolution. As a consequence, current flow is not focal, but distributed across the entire scalp. In Fig 1D, we simulated the electric field in our paradigm. The distribution of current flow is indeed very broad, encompassing several brain areas. Therefore, it is conceivable that our tACS manipulation did not exclusively stimulate the MFC. Due to a complex interplay of brain networks, it remains possible that other brain regions were stimulated as well, potentially interacting or interfering with our RPE effect in declarative learning. Second, tACS only generates weak electrical fields. The simulation in Fig 1D shows that using a stimulation intensity of 2mA caused, at best, an electric field strength of 0.3 V/m, which is on the weak side. The induction of weak electrical fields makes it difficult to entrain endogenous oscillations. This is especially the case if the brain regions that need to be stimulated are located deeper within the brain. For instance [74], reported that low frequency tACS did not modulate ongoing brain activity during resting wakefulness [75]. also found that conventional stimulation parameters are insufficient to induce measurable effects. However, the use of stronger currents might be accompanied by increased discomfort. Third, some researchers raised the issue of brain-state-dependent effects [76–80]. More specifically, tACS effects might depend on the current brain state of the participant. If a participant is in an optimal brain state where brain networks are synchronized enabling high encoding efficiency, stimulating the learning brain might impair learning. If, however, a participant is in a non-optimal brain state where synchronization is less pronounced and accompanied by decreased encoding efficiency, then applying stimulation could facilitate learning and improve memory performance. Importantly [81], have shown that endogenous brain oscillations are entrained only when phase-alignment is achieved between the applied stimulation and the ongoing brain activity (see also [72]).

Therefore, stimulation should ideally be phase-aligned to participants' internal brain states [82]. As we could not measure participants' brain states in our study, it is possible that tACS interacted with ongoing endogenous brain states. Fourth, it remains possible that theta frequency has no effect on RPEs in declarative learning and declarative memory per se. For instance [83], applied theta-frequency (5 Hz) tACS over the ventrolateral prefrontal cortex during the acquisition of face-occupation pairs in older adults. In line with our study, theta-frequency tACS did not affect memory performance. Fifth, due to logistical constraints, a between-subjects design was used. By doing so, individual differences are not easily controlled. This could be mitigated by using a within-subjects design, where each participant is subjected to a real and a sham stimulation condition. Finally, due to the lack of standardized tACS procedures across studies, it remains difficult to draw definitive conclusions. The absence of an effect highlights the importance for understanding its underlying mechanisms [84], and setting up general procedural guidelines with regard to neurostimulation studies [51, 85].

Taken together these issues, we argue that the lack of strong, localized, and phase-dependent stimulation is the most important factor contributing to our null result. Therefore, a follow-up of our study would be to use rhythmic Transcranial Magnetic Stimulation (TMS) to improve spatial resolution and induce stronger electrical fields [86] while simultaneously measuring EEG. Even though the spatial resolution of TMS remains debated [87], it is more focal than tACS. By using a closed-loop approach, brain states are continuously monitored and stimulation can be phase-aligned to individual theta oscillations. As such, we would be in a better position to influence learning. Interestingly, in the same experimental paradigm where rTMS at beta frequency modulated declarative memory [88], tACS at beta frequency did not successfully modulate memory formation [89]. This finding thus further validates the use of (rhythmic) TMS over tACS. To further increase stimulation strength, instead of delivering single pulses at theta frequency, another procedure would be to deliver high-frequency bursts at theta frequency. This procedure has also been shown to increase memory performance and certainty ratings [90, 91] and thus is also a viable alternative for future research.

In summary, the current study examined whether applying theta-frequency (6 Hz) tACS over the MFC modulates the RPE effect in declarative learning. Previous behavioral results were replicated, with SRPEs driving declarative learning. However, theta tACS over the MFC did not modulate the effect of RPEs on declarative learning, and we proposed guidelines for future neuromodulation studies in declarative memory.

## Supporting information

**S1 Fig. Certainty ratings for subjects 1 to 20.**
(TIFF)

**S2 Fig. Certainty ratings for subjects 21 to 41.**
(TIFF)

**S3 Fig. Certainty ratings for subjects 42 to 61.**
(TIFF)

**S4 Fig. Certainty ratings for subjects 62 to 77.**
(TIFF)

**S1 Table. Stimulus material practice set: 6 Dutch words.**
(DOCX)

**S2 Table. Stimulus material practice set: 24 Japanese words.**
(DOCX)

**S3 Table. Stimulus material: 60 Dutch words.**
(DOCX)

**S4 Table. Stimulus material: 240 Swahili words.**
(DOCX)

**S1 File. Sensations questionnaire.**
(DOCX)

## Acknowledgments

We thank Lara Bardi for help with the tACS startup.

## Author Contributions

**Conceptualization:** Kate Ergo, Esther De Loof, Gillian Debra, Tom Verguts.

**Formal analysis:** Kate Ergo, Esther De Loof, Gillian Debra, Bernhard Pastötter.

**Investigation:** Kate Ergo, Esther De Loof, Gillian Debra, Tom Verguts.

**Methodology:** Kate Ergo, Esther De Loof, Gillian Debra, Tom Verguts.

**Resources:** Tom Verguts.

**Supervision:** Tom Verguts.

**Visualization:** Kate Ergo, Esther De Loof, Gillian Debra.

**Writing – original draft:** Kate Ergo, Esther De Loof, Gillian Debra, Bernhard Pastötter, Tom Verguts.

**Writing – review & editing:** Kate Ergo, Esther De Loof, Bernhard Pastötter, Tom Verguts.

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
