## [Decision Letter · Decision Letter 0]

10 Sep 2020

PONE-D-20-23792

Failure to modulate reward prediction errors in declarative learning with theta (6 Hz) frequency transcranial alternating current stimulation.

PLOS ONE

Dear Dr. Ergo,

Thank you for submitting your manuscript to PLOS ONE. After careful consideration, we feel that it has merit but does not fully meet PLOS ONE’s publication criteria as it currently stands. Therefore, we invite you to submit a revised version of the manuscript that addresses the points raised during the review process.

Your manuscript is reviewed by five experts. The reviews are quite mixed, with one rejection and one acceptance. While all reviewers mentioned important points, in particular I would like to strongly encourage authors to expand the Discussion based on the points raised by Reviewer 1 and respond to the points raised by Reviewer 5 in enough details. 

We look forward to receiving your revised manuscript.

Kind regards,

Amir-Homayoun Javadi, PhD

Academic Editor

PLOS ONE

Journal Requirements:

Reviewers' comments:

Reviewer's Responses to Questions

**Comments to the Author**

1. Is the manuscript technically sound, and do the data support the conclusions?

Reviewer #1: Partly

Reviewer #2: Yes

Reviewer #3: Yes

Reviewer #4: Yes

Reviewer #5: Partly

2. Has the statistical analysis been performed appropriately and rigorously? 

Reviewer #1: I Don't Know

Reviewer #2: Yes

Reviewer #3: No

Reviewer #4: Yes

Reviewer #5: Yes

3. Have the authors made all data underlying the findings in their manuscript fully available?

Reviewer #1: No

Reviewer #2: Yes

Reviewer #3: Yes

Reviewer #4: No

Reviewer #5: Yes

4. Is the manuscript presented in an intelligible fashion and written in standard English?

Reviewer #1: Yes

Reviewer #2: Yes

Reviewer #3: Yes

Reviewer #4: Yes

Reviewer #5: Yes

5. Review Comments to the Author

Reviewer #1: Summary of Comments

In this manuscript, Ergo et al. examine the relationship between reward prediction errors (RPE), declarative memory, and tACS-induced theta oscillations. The authors asked participants to learn word pairs with pre-determined RPEs, by delivering rewards solely respective of the choice possibility space and unrelated to ground-truth correctness. Concurrently, they delivered theta-frequency tACS over the medial frontal cortex (MFC) in a subset of subjects. The authors found that, replicating their earlier results, RPEs are positively correlated with memory accuracy – more “surprising” rewards were associated with better declarative memory. However, they found no difference between memory accuracy or the RPE-memory correlation when comparing the stimulation and sham groups.

In its current form, the manuscript appears to meet most – but not all – of the PLoS One criteria for publication. Among the study’s strengths are its specific and well-justified hypothesis, which was reasonably addressed by the experimental manipulation. Additionally, the study authors deserve credit for not overanalyzing or overinterpreting their null results, which are meaningful in that they demonstrate the continued need to better refine and understand noninvasive stimulation methods. The study’s chief weakness relates to PLoS criteria (3), concerning the detail in which experimental results are reported. Data across 76 subjects are only presented in highly summarized form in one figure, making it impossible for readers to assess the underlying variability in the data, its distributional form, the presence of outliers, etc. The use of linear mixed effects models for making inferences does not obviate the need to report data in a meaningful way.

These comments are expanded upon below, in addition to an enumeration of less critical revisions and minor comments.

Major Comments

1. Reporting and visualization of the underlying data. Ergo, et al. assert that SRPEs were correlated with recall performance, and that theta tACS failed to induce any significant change in memory. To that end, one figure (Fig. 2) and the results of several mixed effects models are provided. The authors should be commended for considering mixed-effects models, which account for inter-subject variability in the relationship between SRPE and accuracy. However, this level of data reporting is insufficient for readers to evaluate the analysis or experimental conclusions. It is now fairly standard for manuscripts to plot the distributional form of their data, either as histograms or “swarmplots” overlaid on error bars. From Figure 2, I have no sense of what kind of inter-subject variability is present, whether or not the distributions are normal, and whether or not there are significant outliers. For example, it could be the case that in a subset of subjects, there was a reliable effect of tACS, but that would be impossible to know from the way this data was presented (of course, such findings should not be overinterpreted, but they should be completely obscured either). Moreover, it is also impossible to know the degree of within-subject variability of response choices (e.g. were some subjects always “very certain,” while others made use of the full range of certainty?). This list of possible visualizations is not exhaustive, but generally speaking, the data must be provided with an appropriate amount of granularity – not too much and not too little. In this case, I believe there is too little.

2. Refined hypothesis regarding the effect of tACS on memory. Less critically, but to further improve the manuscript, would be for the authors to offer a mechanistic explanation for exactly how they believe continuous tACS during the acquisition period affected memory performance. As the authors address in their Discussion, there are several factors which could affect the way in which tACS alters memory performance – however, they present a grab-bag of hypothesis without a strong suggestion as to which they feel is most likely. Additionally, none were directly tied to their earlier observation that theta power increased during reward feedback. If this is the case, does it stand to reason that theta tACS should have been delivered selectively during the feedback phase, instead of continuously during the entire acquisition period? (This hypothesis would be most similar to the discussion of brain-state dependent effects.) Moreover, tACS did not appear to reliably decrease memory performance either, a finding which shouldn’t be taken for granted (e.g. Jacobs, et al. 2016 Neuron). Simply put, I think the authors should take a stand on what they believe happened in this experiment.

Minor Comments

1. The caption for Figure 2 is lacking in detail. Error bars should be defined (e.g. +/- 1 SEM) and a brief description of the underlying methods should be provided.

2. How was it decided that the sham stimulation group should receive 40 seconds of stimulation (pg. 11)? Was the stimulation delivered at the very beginning of the task?

3. It would be helpful if the point that subjects do not necessarily learn the correct translation (bottom of pg. 9) be mentioned earlier in the Methods. I was very confused until I read this line!

4. It’s too late to change now, but authors were over-reliant on the word “modulated” in their Introduction, shielding them from making a real prediction about the effect of tACS (last paragraph of pg. 6). Indeed, the authors had just presented evidence that theta phase synchronization “boosts” declarative learning – if this is case, why didn’t they predict that tACS would yield memory improvement? It’s fine if they predicted that tACS would improve memory and it did not, because that’s the way science works. I’m worried that, after seeing the results of the experiment, they wrote their Introduction with the vague use of “modulated memory” to avoid the appearance of being “incorrect” in their initial hypothesis.

5. Line 91: “Implicated” instead of “implied.”

Reviewer #2: The authors are using a paradigm they have previously used to asses the role of RPE in declarative memory. In this instance, they are examining the role of the stimulation to investigate

I am not an expert in EEG so cannot comment in detail on this aspect of the paper but I thought the methods were described clearly for a non-expert.

The task is well suited to answering the question and I was pleased to see both the Bayesian and Frequentist statistics reported.

Overall I felt the paper was very well written and the research question was well motivated. The introduction could cover some additional background on relevance of prediction errors in animal and reinforcement learning. Previously, fMRI has been the dominant tool in this area of research so emphasising the addition of EEG would be nice.

Although the authors have previously found RPE effects using this paradigm I think it would be beneficial to comment on the mixed results/paradigms in this area and note that the results hold within the context of the authors current paradigms.

Reviewer #3: In their work, "Failure to modulate reward prediction errors in declarative learning with theta (6 Hz) frequency transcranial alternating current stimulation," Ergo and colleagues use transcranial alternating current stimulation (tACS) to test for a causal role of theta oscillations in the generating increased declarative memory associated with positively signed reward prediction errors (SRPEs). Using a between-subject design, the authors replicate previous work demonstrating benefits in declarative memory linked to SRPEs. tACS delivered to the MFC did not influence overall memory performance (recognition accuracy), but demonstrated a weak effect on recognition confidence, with subjects in the stimulation group demonstrating lower certainty in recognition decisions. Overall, the manuscript is well written, addresses an interesting and timely research question, and contains reasonable conclusions that are supported by the results. That said, it is unclear how the authors chose the specific stimulation protocol that was used, specifically how it targets medial prefrontal structures implicated in reward processing. In addition, there are some minor statistical issues that could be clarified. I specify these issues in detail below.

Major Issues

1. The authors do a very nice job framing a neural basis for how RPEs modulate memory function in the introduction, describing an established circuit that spans dopaminergic neurons in the midbrain, striatum, hippocampus, and medial frontal cortex (MFC). The main goal of the study was to test whether it was possible to modulate the influence of RPEs on memory via modulation of this circuit by applying tACS to the MFC, with stimulation applied in a bipolar scheme to sponges located over FCz and the neck. To me, there is a bit of a conceptual leap from describing this anatomical network to assuming that the stimulation protocol influences or entrains theta oscillations across the network. Even though event related potential studies commonly identify RPE-related signals at FCz, source modeling and fMRI studies suggest the sources of these signals arise from the striatum and mesolimbic reward structures, including medial frontal cortex (MFC; Carlson et al., 2001, NeuroImage). These medial structures associated with theta are ventrally located, and thus may be difficult to modulate with tACS. Studies using invasive recordings indicate that it is difficult if not impossible to entrain theta oscillations during resting wakefulness (Lafon et al., 2017, Nature Communications). Thus, one simple explanation for the absence of an effect of stimulation is that tACS did modulate or entrain neural activity within the proposed network. It would be useful for the authors to include some form of electric field modeling (e.g., using ROAST, Huang et al., 2019, Journal of Neural Engineering) to show which brain structures would be impacted by stimulation. Discussion of this point could also point out an explanation for the null results.

2. The methodological description for the frequentist statistics used in the manuscript was not detailed enough to understand precisely what was done. From what I can glean, (generalized) linear mixed-effects models were used to test for effects of stimulation and both signed and unsinged RPEs on memory outcomes, as well as their interactions. As the authors report chi-square statistics, I can assume likelihood tests were used to compare full vs. nested models to test for individual effects (e.g., a main effect or interaction), but statistical testing should be described in full in the Methods. The factors included in each model/test should also be detailed. Further, I do not believe it is appropriate to include only random intercept terms in these models. Doing so reduces model generalization by assuming the effects of interest, such as the effect of RPEs on memory (or stimulation on memory), do not vary across individuals (see Barr et al. 2013, Journal of Memory and Language). Random slopes should be included for effects of interest (RPEs, stimulation, etc.) prior to testing on individual model parameters.

3. Is it possible that the effect of tACS on recognition certainty was due to a side effect of stimulation (e.g., reduced attention due to discomfort)? Could this be evaluated with the currently available data? Whereas the authors do a nice job in not overstating this finding, it is practically ignored in the discussion.

4. I found the description of the recognition task to be missing certain details. I assume that subjects were instructed to select the same associate for each Dutch word that was selected in the acquisition phase. However, given the goal of learning the Swahili pairs, without explicit instructions a naïve subject may attempt to select the “correct” pair. Except for nonrewarded trials in the 4-item condition, it would be possible to successfully perform the task in this manner without guessing. The recognition task goal and instructions should be clarified.

5. It is unclear to me whether much of the behavioral results are interpretable as reported (this relates to my point 2, above). At the beginning of the recognition accuracy section (p. 13), the authors report effects of reward and number of options. Is it not the case that these factors are essentially confounded with RPEs in the design? For example, the 1-option condition is always rewarded without a reward prediction error, whereas the magnitude of RPEs increase with more options. Thus, it is unclear to me if the authors are reporting an effect of the number of options or RPEs. If these factors are confounded, it would make sense to only report the effects of signed and unsigned RPEs on memory, given the design.

Minor Issues

1. To improve communication of the design, I think it would be useful to state the proportion of trials in each condition under the design subsection on p.9. It only became obvious to me that the proportion of trials were matched to expectation of reward after viewing Figure 1.

2. On lines 275 and 295, the authors report Bayes factors (BF01) “against the null model.” This language conflicts with standard usage and the description in the Methods on line 236 that BF01 supports the null (in this case of no effect of stimulation).

3. The language on lines 287-288 makes it seem as if there was an effect of stimulation on recognition accuracy. The authors may want to consider changing the language in this section, so it is clear the dependent measure is a measure of recognition certainty or confidence, rather than accuracy.

Reviewer #4: In the article "Failure to modulate reward prediction errors in declarative learning with theta (6 Hz) frequency transcranial alternating current stimulation", Ergo et al. investigated the relationship between signed reward prediction errors (RPEs) in learning trials and subsequent recognition performance as a function of transcranial Alternating Current Stimulation (tACS) at 6Hz applied in the learning phase. Whereas they found a relationship between SRPEs and recognition performance, this relationship was not affected by tACS.

It is always hard to know what to make of these kinds of null findings, but the authors discuss a range of possible reasons why tACS did not show the expected effect on recognition performance. Given that many studies have shown that the timing of feedback is important for the effectiveness of RPEs in modulating learning, I am wondering whether the fact that the phase of the tACS stimulation relative to feedback onset varied might have also contributed to the lack of effect. It is hard to know how to get around this in the current study design given that the time to make a response was not constrained and feedback immediately followed the choice. In theory it might be possible to investigate this question with a post-hoc analysis of the effect of tACS phase at feedback onset, but the number of trials might be too low (and recognition performance too variable) for such an analysis to yield conclusive results. With respect to the variability in recognition performance, it seems that at least one subject might have guessed in a large proportion of the trials, given that the lower end of the range is at or very near the chance level of 25% in many subsets of trials.

I think the clarity of the description of the results could be improved. The experimental design is fairly straight forward, but I found myself having to work a bit at relating the descriptions in the results section to the task. Just to give one example, the first sentence in the "Recognition accuracy" section reads "The data revealed a significant main effect of reward" which the following sentence clarifies. Replacing these first two sentences with something like "Choices which were rewarded during the learning session were more likely to be recognized later (M=64.6% ....) compared to unrewarded choices (M=66.4% ...; stats for main effect)" would require much less effort from the reader.

Minor comment:

There appears to be a typo in Figure 1: the URPE for the unrewarded choice in the 4 options condition should be 0.25, not 0.75. The version of the figure that was included in the manuscript was quite degraded --- this can usually be avoided by uploading figures in vector graphic formats such as SVG instead of in bitmap formats such as tiff.

Reviewer #5: Ergo et al present a study using 6 Hz tACS with the goal of modulating the neural responses to reward prediction errors (RPEs) during a declarative memory task. The authors test a stimulation and a sham group in a paired associates learning task with reward feedback. The authors find that RPE is related to memory performance, but do not find that 6 Hz stimulation modulates the effect of RPE on memory. The authors report some evidence that stimulation affects Certainty ratings (memory meta-judgements). The manuscript addresses an important question about whether and how RPEs affect declarative memory. The use of stimulation as to causally manipulate the effects is important and it is beneficial for the community to be aware of this null result. However, I have fundamental questions about whether the data support the authors’ theoretical model of RPEs to begin with.

Major comments:

1) The Results report that the Unrewarded condition showed higher accuracy than the Rewarded condition (66.4% vs 64.6%) but it is not mentioned whether this difference was significant. This seems to be a surprising effect, given the literature reviewed in the Introduction and Discussion that associates positive RPEs with better declarative memory.

2) Another question about the benefit for Unrewarded vs. Rewarded trials--I’m confused as to how this can be the case if it is also true that accuracy linearly increased with SRPE. The negative SRPEs [-0.5, -0.25] both came from the Unrewarded condition, so if Unrewarded accuracy is higher than Rewarded accuracy, which includes SRPEs [0, 0.5 and 0.75], then wouldn’t this be evidence in favor of a U-shaped effect of RPE instead of the linear effect that the authors argue, which would contradict their model of RPE?

3) Although the between groups analyses are based on relatively large samples, the number of trials in each condition for within-subject analyses is small, especially given the need to adjudicate between the U-shaped and linear RPE models. I’m not sure how you get around this without altering the study design, except perhaps to collapse the data into negative, zero, and positive SRPE.

4) The difference in perception of discomfort duration between the stimulation and sham groups was significant and a potential confound, but the mean ratings are not reported. What were the actual mean duration numbers for each group? For the significant effects of stimulation that are reported (for Certainty ratings) the authors should bootstrap a subset of stimulation participants that are matched with the Sham group on this and all other questionnaire/demographic variables to confirm that discomfort duration is not confounding these effects (alternatively they could regress out the estimated discomfort duration).

Minor comment:

There are a few instances (e.g. Pages 9 & 11) where the manuscript refers to the placement of the reference electrode as being “in the neck.” Perhaps consider changing this to “on the neck.”

6. PLOS authors have the option to publish the peer review history of their article (what does this mean?). If published, this will include your full peer review and any attached files.

Reviewer #1: No

Reviewer #2: No

Reviewer #3: No

Reviewer #4: No

Reviewer #5: No

---

## [Author Response · Author response to Decision Letter 0]

16 Oct 2020

Response to Reviewers

Dear Dr. Javadi,

We thank you and the five reviewers for the thorough evaluation of our paper titled “Failure to modulate reward prediction errors in declarative learning with theta (6 Hz) frequency transcranial alternating current stimulation”. We have carefully reviewed the comments and have revised the manuscript accordingly. We think the manuscript has improved as a result. Below you can find our answers to the reviewers’ comments made to which we responded in a point-by-point fashion. 

Sincerely,

Kate Ergo, Esther De Loof, Gillian Debra, Bernhard Pastötter, and Tom Verguts

Reviewer 1

Comment: Reporting and visualization of the underlying data. Ergo, et al. assert that SRPEs were correlated with recall performance, and that theta tACS failed to induce any significant change in memory. To that end, one figure (Fig. 2) and the results of several mixed effects models are provided. The authors should be commended for considering mixed-effects models, which account for inter-subject variability in the relationship between SRPE and accuracy. However, this level of data reporting is insufficient for readers to evaluate the analysis or experimental conclusions. It is now fairly standard for manuscripts to plot the distributional form of their data, either as histograms or “swarmplots” overlaid on error bars. From Figure 2, I have no sense of what kind of inter-subject variability is present, whether or not the distributions are normal, and whether or not there are significant outliers. For example, it could be the case that in a subset of subjects, there was a reliable effect of tACS, but that would be impossible to know from the way this data was presented (of course, such findings should not be overinterpreted, but they should be completely obscured either). Moreover, it is also impossible to know the degree of within-subject variability of response choices (e.g. were some subjects always “very certain,” while others made use of the full range of certainty?). This list of possible visualizations is not exhaustive, but generally speaking, the data must be provided with an appropriate amount of granularity – not too much and not too little. In this case, I believe there is too little.

Response: We agree with reviewer 1 that adding more information with regard to the distribution of the data facilitates the interpretation of the current results. We have now overlaid the average recognition accuracies and certainty ratings with individual data points representing mean accuracy (gray dots) and certainty (gray dots for correct recognitions and gray triangles for incorrect recognitions) per condition for each participant (swarmplots). We also added to the description of Figure 2 that recognition accuracies and certainty ratings are shown together with their 95% confidence interval. 

Comment: Refined hypothesis regarding the effect of tACS on memory. Less critically, but to further improve the manuscript, would be for the authors to offer a mechanistic explanation for exactly how they believe continuous tACS during the acquisition period affected memory performance. As the authors address in their Discussion, there are several factors which could affect the way in which tACS alters memory performance – however, they present a grab-bag of hypothesis without a strong suggestion as to which they feel is most likely. Additionally, none were directly tied to their earlier observation that theta power increased during reward feedback. If this is the case, does it stand to reason that theta tACS should have been delivered selectively during the feedback phase, instead of continuously during the entire acquisition period? (This hypothesis would be most similar to the discussion of brain-state dependent effects.) Moreover, tACS did not appear to reliably decrease memory performance either, a finding which shouldn’t be taken for granted (e.g. Jacobs, et al. 2016 Neuron). Simply put, I think the authors should take a stand on what they believe happened in this experiment.

Response: Our theory holds that RPEs cause theta oscillations, and (midfrontal) theta oscillations improve ongoing learning in the brain (see also [1], for a discussion of this theory). From this perspective, we predicted both an effect of (theta-frequency) stimulation as well as an interaction between SRPE and stimulation on declarative learning. However, in hindsight, we believe that a stronger theta stimulation would be required for a robust effect. As a consequence, in the current version of the paper, we propose (r)TMS as a viable alternative to induce theta oscillations. The main reason is that TMS is known to cause highly localized (here, to MFC) and supra-threshold electric fields, as opposed to tACS. 

However, brain-state-dependency is another important factor that must not be overlooked. In our experiment, we were unable to simultaneously measure EEG signals, making it impossible to know what phase (i.e., brain-state) participants were in. Therefore, an interesting follow-up study would be to measure EEG while concurrently applying (r)TMS. One major advantage of this approach is that it allows stimulation to be phase-aligned with endogenous brain activity. However, tACS induces several artifacts in the EEG signal that have to be eliminated first. One way to achieve this is by using the method of [2] in which principal component analysis (PCA) is used to filter out stimulation artifacts. In the manuscript, we now take a clearer stance on what we believe to be the most likely explanation for our null finding. We added two papers on brain-state dependent effects emphasizing the importance of initial brain states [3] and phase-alignment [4].

Thus, based on the theory and the current experimental design and results, we now explain what we think was the major problem in the design; and we present a specific alternative procedure in the General Discussion for future research.

Comment: The caption for Figure 2 is lacking in detail. Error bars should be defined (e.g. +/- 1 SEM) and a brief description of the underlying methods should be provided.

Response: We thank reviewer 1 for pointing out the lack of detail in Figure 2. We now stated more clearly that the average recognition accuracies and certainty ratings are plotted with their 95% confidence intervals. We also added individual-subject data points (averaged per condition) that show the distribution of the data. More specifically, we overlaid the average recognition accuracies and certainty ratings with individual data points representing mean accuracies (gray dots) and certainties per condition (gray dots for correct recognitions and gray triangles for incorrect recognitions) for each participant.

Comment: How was it decided that the sham stimulation group should receive 40 seconds of stimulation (pg. 11)? Was the stimulation delivered at the very beginning of the task?

Response: We now added to the paper that the sham stimulation was administered at the beginning of the acquisition task for a duration of 40 seconds. Stimulation duration in the sham stimulation group was based on an study using tDCS [5]. We decided to keep stimulation time short to avoid actually stimulating the brain by inducing changes in cortical excitability [6]. 

Comment: It would be helpful if the point that subjects do not necessarily learn the correct translation (bottom of pg. 9) be mentioned earlier in the Methods. I was very confused until I read this line!

Response: We clarified this by explicitly stating in the Introduction (page 3) and the Methods (page 8) sections that by fixing the number of eligible Swahili translations and whether a trial was rewarded or not, participants did not learn the actual Swahili translations for the Dutch words. 

Comment: It’s too late to change now, but authors were over-reliant on the word “modulated” in their Introduction, shielding them from making a real prediction about the effect of tACS (last paragraph of pg. 6). Indeed, the authors had just presented evidence that theta phase synchronization “boosts” declarative learning – if this is case, why didn’t they predict that tACS would yield memory improvement? It’s fine if they predicted that tACS would improve memory and it did not, because that’s the way science works. I’m worried that, after seeing the results of the experiment, they wrote their Introduction with the vague use of “modulated memory” to avoid the appearance of being “incorrect” in their initial hypothesis.

Response: We apologize if by using the word “modulated”, our predictions were unclear. In the revised version of the manuscript, we have tried making our hypotheses more explicit. We would also like to point out that we always intended to investigate the modulation of RPEs (and their effect) on declarative learning by means of 6 Hz tACS administered over the MFC. More specifically, we believe that if declarative learning is driven by theta frequency oscillations in MFC (see response to earlier comment also), then subsequent recognition accuracies and certainty ratings should be modulated by tACS that is delivered during RPE computation by the individual. Recognition accuracies and certainty ratings should be higher in the real compared to sham stimulation group. 

Comment: Line 91: “Implicated” instead of “implied.”

Response: We changed the word “implied” to “implicated”.

Reviewer 2

Comment: Overall I felt the paper was very well written and the research question was well motivated. The introduction could cover some additional background on relevance of prediction errors in animal and reinforcement learning. Previously, fMRI has been the dominant tool in this area of research so emphasising the addition of EEG would be nice.

Response: We thank the reviewer for the positive evaluation. We now added the paper of [7] on the role of RPEs in procedural learning to the Introduction. We also want to make clear that the current paper uses tACS as a neurostimulation technique without concurrently measuring any brain signals, such as EEG. 

Comment: Although the authors have previously found RPE effects using this paradigm I think it would be beneficial to comment on the mixed results/paradigms in this area and note that the results hold within the context of the authors current paradigms.

Response: It is correct that studies investigating the effect of RPEs in declarative learning have sometimes led to contradictory results (for a review, see [8]). Some studies found an SRPE effect, whereas others found an URPE effect. Whereas most studies show a benefit of RPEs, one particular study found a decrease in memory performance. In the second paragraph of the Discussion section we now briefly talk about how the current results relate to earlier findings. 

Reviewer 3

Comment: The authors do a very nice job framing a neural basis for how RPEs modulate memory function in the introduction, describing an established circuit that spans dopaminergic neurons in the midbrain, striatum, hippocampus, and medial frontal cortex (MFC). The main goal of the study was to test whether it was possible to modulate the influence of RPEs on memory via modulation of this circuit by applying tACS to the MFC, with stimulation applied in a bipolar scheme to sponges located over FCz and the neck. To me, there is a bit of a conceptual leap from describing this anatomical network to assuming that the stimulation protocol influences or entrains theta oscillations across the network. Even though event related potential studies commonly identify RPE-related signals at FCz, source modeling and fMRI studies suggest the sources of these signals arise from the striatum and mesolimbic reward structures, including medial frontal cortex (MFC; Carlson et al., 2001, NeuroImage). These medial structures associated with theta are ventrally located, and thus may be difficult to modulate with tACS. Studies using invasive recordings indicate that it is difficult if not impossible to entrain theta oscillations during resting wakefulness (Lafon et al., 2017, Nature Communications). Thus, one simple explanation for the absence of an effect of stimulation is that tACS did modulate or entrain neural activity within the proposed network. It would be useful for the authors to include some form of electric field modeling (e.g., using ROAST, Huang et al., 2019, Journal of Neural Engineering) to show which brain structures would be impacted by stimulation. Discussion of this point could also point out an explanation for the null results.

Response: We agree that the spatial resolution of tACS is rather limited and that this has two major consequences: First, current flow is not focal and distributed across the scalp. This means that we might be stimulating other brain networks that we did not intend to stimulate. This might have led to unwanted interactions. Second, it remains possible that our stimulation intensity of 2mA is insufficient to induce electrical fields that are strong enough to entrain ongoing brain oscillation. In our discussion of the limitations of the current study, we further substantiated this by adding the papers of [9] and [10]. To have a better idea how the electric field looks like in our experiment, we used ROAST [11] for a simulation. The result of this simulation was added to Figure 1D. From Figure 1D it can be seen that our tACS stimulation was not focal and only introduced weak electrical fields (maximal electric field strength of 0.3 V/m). As a result, we believe that our future studies would benefit from using (r)TMS instead of tACS, as this method allows for stronger and more focal stimulation. 

For full transparency, the source code we used is reported here:

roast([], ...

 {'FCz', 2, 'Nk2', -2}, ...

 'electype', {'pad', 'pad'}, ...

 'elecsize', {[65, 50, 3], [65, 50, 3]}, ...

 'elecori', {'ap', 'ap'}, ...

 'zeropadding', 110, ...

 'simulationTag', 'tACS_simulation');

Comment: The methodological description for the frequentist statistics used in the manuscript was not detailed enough to understand precisely what was done. From what I can glean, (generalized) linear mixed-effects models were used to test for effects of stimulation and both signed and unsinged RPEs on memory outcomes, as well as their interactions. As the authors report chi-square statistics, I can assume likelihood tests were used to compare full vs. nested models to test for individual effects (e.g., a main effect or interaction), but statistical testing should be described in full in the Methods. The factors included in each model/test should also be detailed. Further, I do not believe it is appropriate to include only random intercept terms in these models. Doing so reduces model generalization by assuming the effects of interest, such as the effect of RPEs on memory (or stimulation on memory), do not vary across individuals (see Barr et al. 2013, Journal of Memory and Language). Random slopes should be included for effects of interest (RPEs, stimulation, etc.) prior to testing on individual model parameters.

Response: We agree that not taking into account the random slopes can increase the risk of Type I errors in LMEs. However, we want to point out that during our analyses, we used a bottom-up modeling approach to verify the validity of adding random slopes. This approach allows us to leave out insignificant random slopes from the start. In particular, we checked whether the models without random slopes gives us similar p-values for the fixed effects. If so, we could drop the random slopes. This makes scripts more readable, faster to execute and less complex to describe.

Relatedly, adding random slopes to the model takes up a lot of statistical power. Given the limited number of trials in each cell of the design (e.g., 5 trials in the four-option rewarded condition), this (traditional) approach is usually suboptimal in our case. Nevertheless, in response to the reviewer, we reran the analyses for recognition accuracy and certainty ratings including random slopes for all factors (i.e., accuracy, SRPE and stimulation). Reassuringly, the conclusions for the main variable of interest, recognition accuracy, remain the same with or without random slopes. However, the model for certainty ratings failed to converge. Although output was provided for the model, these parameter estimates cannot be trusted. We therefore decided to not include random slopes in our analyses and only report the fixed effects and random intercept models in the manuscript. We added some more details to the Data analysis section with regard to our statistical analyses, e.g., “We report the �² statistics from the ANOVA Type III tests.”. For full transparency, we reproduce the code and model output below.

Analysis code and output:

1) Models without random slopes:

a) Accuracy (ACC): 

fit = glmer(ACC ~ (1|Subject) + SRPE * Stimulation, data, family = binomial); Anova(fit, type = "III"); summary(fit)

 Chisq Df Pr(>Chisq) 

(Intercept) 81.0862 1 < 2.2e-16 ***

SRPE 9.1268 1 0.002519 ** 

Stimulation 1.4204 1 0.233338 

SRPE:Stimulation 0.0035 1 0.953096

b) Certainty:

fit = lmer(Certainty ~ (1|Subject) + ACC * SRPE * Stimulation, data); Anova(fit, type="III"); summary(fit) 

 Chisq Df Pr(>Chisq) 

(Intercept) 2368.1130 1 < 2.2e-16 ***

ACC 1169.5293 1 < 2.2e-16 ***

SRPE 1.3428 1 0.246543 

Stimulation 0.3035 1 0.581680 

ACC:SRPE 7.6293 1 0.005743 ** 

ACC:Stimulation 5.3692 1 0.020495 * 

SRPE:Stimulation 1.6111 1 0.204343 

ACC:SRPE:Stimulation 0.5436 1 0.460927 

2) Models with random slopes for all factors of interest:

a) Accuracy (ACC): 

fit = glmer(ACC ~ (1+SRPE+Stimulation|Subject) + SRPE * Stimulation, data, family = binomial); Anova(fit, type = "III"); summary(fit) 

 Chisq Df Pr(>Chisq) 

(Intercept) 81.2406 1 < 2.2e-16 ***

SRPE 7.5006 1 0.006168 ** 

Stimulation 1.5406 1 0.214524 

SRPE:Stimulation 0.1247 1 0.723957 

b) Certainty: 

fit = lmer(Certainty ~ (1+ACC+SRPE+Stimulation|Subject) + ACC * SRPE * Stimulation, data); Anova(fit, type = "III"); summary(fit) 

 Chisq Df Pr(>Chisq) 

(Intercept) 1885.6212 1 < 2.2e-16 ***

ACC 502.3692 1 < 2.2e-16 ***

SRPE 1.4988 1 0.220853 

Stimulation 0.2137 1 0.643879 

ACC:SRPE 8.2541 1 0.004066 ** 

ACC:Stimulation 2.4995 1 0.113885 

SRPE:Stimulation 1.5487 1 0.213332 

ACC:SRPE:Stimulation 0.6585 1 0.417095 

Comment: Is it possible that the effect of tACS on recognition certainty was due to a side effect of stimulation (e.g., reduced attention due to discomfort)? Could this be evaluated with the currently available data? Whereas the authors do a nice job in not overstating this finding, it is practically ignored in the discussion.

Response: We thank reviewer 3 for pointing out this potential confound in the data. To answer this question, we ran an additional analysis. Although participants in the real stimulation group reported increased discomfort duration, the effect of discomfort duration did not significantly affect certainty in the real (χ2(1, N = 31) = 0.93, p = .33) and sham (χ2(1, N = 30) = 0.19, p = .66) stimulation groups. This suggests that the effect of stimulation on certainty is not due to discomfort. However, as the reviewer notes, we chose not to (over)interpret this finding.

Comment: I found the description of the recognition task to be missing certain details. I assume that subjects were instructed to select the same associate for each Dutch word that was selected in the acquisition phase. However, given the goal of learning the Swahili pairs, without explicit instructions a naïve subject may attempt to select the “correct” pair. Except for nonrewarded trials in the 4-item condition, it would be possible to successfully perform the task in this manner without guessing. The recognition task goal and instructions should be clarified.

Response: We apologize if the description of the recognition task was unclear. In the recognition task, participants had to select for each Dutch word, the correct (or “to-be-learned”) Swahili translation. Thus, participants did not have to select the same Swahili translation twice per se (once in the acquisition task and once in the recognition test). We would also like to add that although this was indeed not explicitly communicated, the goal of the experiment was very obvious to the participants. More specifically, on each trial, there was a clear, normatively correct choice that had to be remembered. Furthermore, the colors (i.e., red/green) and the feedback (i.e., wrong/correct) also clearly indicated to the participants what the intention was of the experiment. Also, none of the subjects reported any confusion about the goal of the experiment.

In the revised version of the paper, we added more details to the description of the recognition task. We now state that: “In the recognition task, participants’ recognition was tested on 60 Dutch-Swahili word pairs that were acquired during the acquisition task” and “Note also that although not explicitly communicated to the participants, there was a clear, normatively correct choice that had to be remembered on each trial. The intention of the experiment was also made clear by the colors (i.e., red/green) and the feedback (i.e., wrong/correct) that were used in the acquisition task.”.

Comment: It is unclear to me whether much of the behavioral results are interpretable as reported (this relates to my point 2, above). At the beginning of the recognition accuracy section (p. 13), the authors report effects of reward and number of options. Is it not the case that these factors are essentially confounded with RPEs in the design? For example, the 1-option condition is always rewarded without a reward prediction error, whereas the magnitude of RPEs increase with more options. Thus, it is unclear to me if the authors are reporting an effect of the number of options or RPEs. If these factors are confounded, it would make sense to only report the effects of signed and unsigned RPEs on memory, given the design.

Response: We agree with reviewer 3 that reporting the effects of reward and number of options is confusing and unnecessary. In our experiment, RPEs are computed by subtracting the number of options from reward. Therefore, the RPE-effect combines reward and number of options in a theoretically motivated way. For the sake of clarity, we now only included the effects of signed and unsigned RPEs on recognition accuracy and certainty. 

Comment: To improve communication of the design, I think it would be useful to state the proportion of trials in each condition under the design subsection on p.9. It only became obvious to me that the proportion of trials were matched to expectation of reward after viewing Figure 1.

Response: In the revised manuscript, we now state that: “In addition, the proportion of trials in each cell of the design were matched to reward expectation (i.e., 100% rewarded trials in the one-option condition, 50% rewarded trials in the two-option condition and 25% rewarded trials in the four-option condition).”

Comment: On lines 275 and 295, the authors report Bayes factors (BF01) “against the null model.” This language conflicts with standard usage and the description in the Methods on line 236 that BF01 supports the null (in this case of no effect of stimulation).

Response: We agree that the language used to describe BF01 was confusing. We now explicitly state for each BF that BF_ab indicates evidence for model a compared to model b.

Comment: The language on lines 287-288 makes it seem as if there was an effect of stimulation on recognition accuracy. The authors may want to consider changing the language in this section, so it is clear the dependent measure is a measure of recognition certainty or confidence, rather than accuracy.

Response: To make it clearer that we are talking about certainty, we changed the sentence to the following: “In addition, the data revealed a significant interaction between stimulation and recognition accuracy on certainty ratings, χ2(1, N = 76) = 5.37, p = .02.”.

Reviewer 4

Comment: It is always hard to know what to make of these kinds of null findings, but the authors discuss a range of possible reasons why tACS did not show the expected effect on recognition performance. Given that many studies have shown that the timing of feedback is important for the effectiveness of RPEs in modulating learning, I am wondering whether the fact that the phase of the tACS stimulation relative to feedback onset varied might have also contributed to the lack of effect. It is hard to know how to get around this in the current study design given that the time to make a response was not constrained and feedback immediately followed the choice. In theory it might be possible to investigate this question with a post-hoc analysis of the effect of tACS phase at feedback onset, but the number of trials might be too low (and recognition performance too variable) for such an analysis to yield conclusive results. With respect to the variability in recognition performance, it seems that at least one subject might have guessed in a large proportion of the trials, given that the lower end of the range is at or very near the chance level of 25% in many subsets of trials.

Response: We agree with reviewer 4 that the phase of tACS stimulation might be an important factor leading to our null result. However, at the time of data collection, our infrastructure did not allow us to concurrently measure EEG signals while administering theta-frequency tACS. We acknowledge that this is a shortcoming of the current experiment that should be controlled in the future. In the Discussion section of the manuscript, we now clearly state that we believe that the lack of phase-dependent stimulation is an important factor contributing to our null result. We also propose a viable alternative for a future follow-up study using rTMS while simultaneously measuring EEG.

Comment: I think the clarity of the description of the results could be improved. The experimental design is fairly straight forward, but I found myself having to work a bit at relating the descriptions in the results section to the task. Just to give one example, the first sentence in the "Recognition accuracy" section reads "The data revealed a significant main effect of reward" which the following sentence clarifies. Replacing these first two sentences with something like "Choices which were rewarded during the learning session were more likely to be recognized later (M=64.6% ....) compared to unrewarded choices (M=66.4% ...; stats for main effect)" would require much less effort from the reader.

Response: We rewrote some sentences in the Results section and hope that by doing so this has facilitated relating the task description with its results.

Comment: There appears to be a typo in Figure 1: the URPE for the unrewarded choice in the 4 options condition should be 0.25, not 0.75. The version of the figure that was included in the manuscript was quite degraded --- this can usually be avoided by uploading figures in vector graphic formats such as SVG instead of in bitmap formats such as tiff.

Response: We thank reviewer 4 for pointing out the typo in Figure 1 and have now uploaded a corrected version of the figure. The low figure quality is likely due to the conversion to PDF; we have increased the resolution of the figures, and will make sure to upload high-quality figures upon acceptance of the paper.

Reviewer 5

Comment: The Results report that the Unrewarded condition showed higher accuracy than the Rewarded condition (66.4% vs 64.6%) but it is not mentioned whether this difference was significant. This seems to be a surprising effect, given the literature reviewed in the Introduction and Discussion that associates positive RPEs with better declarative memory.

Response: There was a significant main effect of reward on recognition accuracy. Contrary to our expectation and to our previous studies, this effect was in the opposite direction with lower accuracy on rewarded trials (64.6%) compared to non-rewarded trials (66.4%). We do however want to point out that in all our previous studies [1,12], we found a significant positive effect of reward, with rewarded word pairs being better remembered than non-rewarded word pairs. This means that in the current study, the SRPE-effect is indeed mainly driven by the number of options. In addition, the one-option condition (i.e., RPE = 0) is consistently associated with a lower accuracy, therefore causing the unrewarded trials to be better remembered than the rewarded trials. This was again the case in the current experiment. Running the analyses without the one-option condition revealed significantly higher accuracies for rewarded trials (M = 70.2%, SD = 15.7%, range = 27%�100%) compared to unrewarded trials (M = 66.4%, SD = 15.8%, range = 32%�100%), χ2(1, N = 76) = 9.47, p = .002. However, in response to Reviewer 3, we removed the reward and number of options factors from the analyses, and only report the effect of SRPE.

Comment: Another question about the benefit for Unrewarded vs. Rewarded trials--I’m confused as to how this can be the case if it is also true that accuracy linearly increased with SRPE. The negative SRPEs [-0.5, -0.25] both came from the Unrewarded condition, so if Unrewarded accuracy is higher than Rewarded accuracy, which includes SRPEs [0, 0.5 and 0.75], then wouldn’t this be evidence in favor of a U-shaped effect of RPE instead of the linear effect that the authors argue, which would contradict their model of RPE?

Response: It is true that in the current study, unrewarded trials are associated with (slightly) increased memory performance compared to rewarded trials. However, we do not agree that the effect fits an URPE-pattern. From Figure 2 it can be seen that the slopes are positive when comparing the 2-options versus 4-options trials, for both rewarded and unrewarded trials. With an URPE pattern, one would expect the slope to be negative between the 2-options and 4-options condition for unrewarded trials, but positive for rewarded trials. 

Comment: Although the between groups analyses are based on relatively large samples, the number of trials in each condition for within-subject analyses is small, especially given the need to adjudicate between the U-shaped and linear RPE models. I’m not sure how you get around this without altering the study design, except perhaps to collapse the data into negative, zero, and positive SRPE.

Response: We agree with reviewer 5 that the number of trials in certain cells of the design are rather low. In our more recent studies (e.g., [1]), we have increased the number of trials in each cell of the design. This was done by no longer assigning the number of trials in proportion to the number of conditions (i.e., 1/3 trials in the one-option condition, 1/3 trials in the two-option condition and 1/3 trials in the four-option condition). Specifically, in this novel design, there were more trials in the 4-option condition, so that we could increase the number of data in the (least frequent) “reward, 4 option” cell of the design.

Comment: The difference in perception of discomfort duration between the stimulation and sham groups was significant and a potential confound, but the mean ratings are not reported. What were the actual mean duration numbers for each group? For the significant effects of stimulation that are reported (for Certainty ratings) the authors should bootstrap a subset of stimulation participants that are matched with the Sham group on this and all other questionnaire/demographic variables to confirm that discomfort duration is not confounding these effects (alternatively they could regress out the estimated discomfort duration).

Response: We thank reviewer 5 for pointing out this potential confound in the data. To answer this question, we ran an additional analysis. Although participants in the real stimulation group reported increased discomfort duration, the effect of discomfort duration did not significantly affect certainty rating in the real (χ2(1, N = 31) = 0.93, p = .33) and sham (χ2(1, N = 30) = 0.19, p = .66) stimulation groups. This suggests that discomfort in itself does not change the certainty ratings. Additionally, we added the mean ratings, standard deviation and range for the following three questions: (1) When the discomfort began, (2) How much these sensations affected their performance, and (3) How long the discomfort lasted. 

We have changed the paragraph to the following: “Furthermore, there were no significant differences between stimulation groups with regard to when the discomfort began, t(58.90) = 0.48, p = .63 (real: M = 1.23, SD = 0.50, range = 0�2; sham: M = 1.17 , SD = 0.46 , range = 0�2), and how much these sensations affected their performance, t(53.77) = 1.13, p = .26 (real: M = 1.39, SD = 0.62, range = 0�4; sham: M = 1.23 , SD = 0.43, range = 0�4). Participants in the real stimulation group did report that the discomfort lasted significantly longer compared to the sham stimulation group, t(40.33) = 3.35, p = .002 (real: M = 1.68, SD = 0.83, range = 0�2; sham: M = 1.13 , SD = 0.35, range = 0�2).”

Comment: There are a few instances (e.g. Pages 9 & 11) where the manuscript refers to the placement of the reference electrode as being “in the neck.” Perhaps consider changing this to “on the neck.”

Response: We replaced all instances where we mentioned “in the neck” to “on the neck”. 

References

1. Calderon CB, De Loof E, Ergo K, Snoeck A, Boehler CN, Verguts T. Signed reward prediction errors in the ventral striatum drive episodic memory. bioRxiv. 2020 Jan 3;2020.01.03.893578. 

2. Guarnieri R, Brancucci A, D’Anselmo A, Manippa V, Swinnen SP, Tecchio F, et al. A computationally efficient method for the attenuation of alternating current stimulation artifacts in electroencephalographic recordings. J Neural Eng [Internet]. 2020 Aug 17 [cited 2020 Sep 16];17(4):046038. Available from: https://doi.org/10.1088/1741-2552/aba99d

3. Silvanto J, Muggleton N, Walsh V. State-dependency in brain stimulation studies of perception and cognition. Trends Cogn Sci. 2008 Dec 1;12(12):447–54. 

4. Thut G, Schyns PG, Gross J. Entrainment of perceptually relevant brain oscillations by non-invasive rhythmic stimulation of the human brain. Front Psychol [Internet]. 2011 Jul 20 [cited 2020 Sep 25];2(JUL):170. Available from: http://journal.frontiersin.org/article/10.3389/fpsyg.2011.00170/abstract

5. Gandiga PC, Hummel FC, Cohen LG. Transcranial DC stimulation (tDCS): a tool for double-blind sham-controlled clinical studies in brain stimulation. Clin Neurophysiol. 2006;117(4):845–50. 

6. Nitsche MA, Cohen LG, Wassermann EM, Priori A, Lang N, Antal A, et al. Transcranial direct current stimulation: State of the art 2008. Brain Stimul. 2008;1(3):206–23. 

7. Schultz W, Dayan P, Montague PR. A neural substrate of prediction and reward. Science (80- ). 1997;275(5306):1593–9. 

8. Ergo K, De Loof E, Verguts T. Reward prediction error and declarative memory. Trends Cogn Sci. 2020;24(5):388–97. 

9. Lafon B, Henin S, Huang Y, Friedman D, Melloni L, Thesen T, et al. Low frequency transcranial electrical stimulation does not entrain sleep rhythms measured by human intracranial recordings. Nat Commun [Internet]. 2017 Dec 1 [cited 2020 Sep 21];8(1):1–14. Available from: www.nature.com/naturecommunications

10. Vöröslakos M, Takeuchi Y, Brinyiczki K, Zombori T, Oliva A, Fernández-Ruiz A, et al. Direct effects of transcranial electric stimulation on brain circuits in rats and humans. Nat Commun [Internet]. 2018 Dec 1 [cited 2020 Sep 23];9(1):1–17. Available from: www.nature.com/naturecommunications

11. Huang Y, Datta A, Bikson M, Parra LC. Realistic volumetric-approach to simulate transcranial electric stimulation-ROAST-a fully automated open-source pipeline. J Neural Eng. 2019;16(5). 

12. De Loof E, Ergo K, Naert L, Janssens C, Talsma D, Van Opstal F, et al. Signed reward prediction errors drive declarative learning. Ito E, editor. PLoS One. 2018 Jan;13(1):e0189212.

---

## [Decision Letter · Decision Letter 1]

9 Nov 2020

PONE-D-20-23792R1

Failure to modulate reward prediction errors in declarative learning with theta (6 Hz) frequency transcranial alternating current stimulation.

PLOS ONE

Dear Dr. Ergo,

Thank you for submitting your manuscript to PLOS ONE. After careful consideration, we feel that it has merit but does not fully meet PLOS ONE’s publication criteria as it currently stands. Therefore, we invite you to submit a revised version of the manuscript that addresses the points raised during the review process.

All reviewers suggested Acceptance and I am also happy with the current version of the document. Only, please add/amend the figures based on Reviewer 1's comments. The additional figures can go in the Supplementary part. Furthermore, currently your data is stored on OSF, but requires permission to access. I would suggest you, if possible, to make it public so that everybody can access your data. I will make the final decision on reception of your new submission with no further review.

We look forward to receiving your revised manuscript.

Kind regards,

Amir-Homayoun Javadi, PhD

Academic Editor

PLOS ONE

Reviewers' comments:

Reviewer's Responses to Questions

**Comments to the Author**

1. If the authors have adequately addressed your comments raised in a previous round of review and you feel that this manuscript is now acceptable for publication, you may indicate that here to bypass the “Comments to the Author” section, enter your conflict of interest statement in the “Confidential to Editor” section, and submit your "Accept" recommendation.

Reviewer #1: (No Response)

Reviewer #3: All comments have been addressed

Reviewer #4: All comments have been addressed

Reviewer #5: All comments have been addressed

2. Is the manuscript technically sound, and do the data support the conclusions?

Reviewer #1: Yes

Reviewer #3: Yes

Reviewer #4: (No Response)

Reviewer #5: Yes

3. Has the statistical analysis been performed appropriately and rigorously? 

Reviewer #1: Yes

Reviewer #3: Yes

Reviewer #4: (No Response)

Reviewer #5: Yes

4. Have the authors made all data underlying the findings in their manuscript fully available?

Reviewer #1: No

Reviewer #3: Yes

Reviewer #4: (No Response)

Reviewer #5: Yes

5. Is the manuscript presented in an intelligible fashion and written in standard English?

Reviewer #1: Yes

Reviewer #3: Yes

Reviewer #4: (No Response)

Reviewer #5: Yes

6. Review Comments to the Author

Reviewer #1: In my initial review, I requested that the authors revise and resubmit their manuscript while addressing several key concerns: (1) a more detailed reporting of experimental results and methodology, and (2) a need to more directly address why their study failed to find an effect, and to pose a specific hypothesis for testing in a follow-up study.

I commend the authors on taking these critiques to heart, as well as addressing several additional concerns raised by other reviewers. Importantly, they have substantially revised their Discussion and now more precisely delineate how brain-state-dependent effects may have driven their null result, offering the framework of a new experimental design to address this question in future work.

Additionally, I appreciate that the distributional form of their data has been included in their figures, though I would recommend creating a clearer visual distinction in the data points in Fig. 2 panels (C) and (D), to separate incorrect vs. correct recognition (I believe they are current using triangle/circle markers, which are difficult to visually distinguish – the authors could consider jittering the vertical alignment to create more distinct columns of data points).

As I mentioned in my original review, the authors certainly have the flexibility to present additional data that can clarify or support their results. For example, within-subject behavioral responses could be presented that demonstrate the extent to which certain subjects may or may not have used the full dynamic range of certainty ratings. Additionally, the authors could present a visualization that essentially captures the purpose of their LME, by showing the per-subject relationship between SRPE and certainty/accuracy. However, I would leave the need for such changes at the discretion of the editor.

In summary, I believe the authors have adequately addressed my concerns and that this manuscript is suitable for publication in PLOS ONE. Nonetheless, I would still encourage the authors to adopt a higher standard of clarity and transparency in the presentation of their data.

Reviewer #3: The authors have addressed all my concerns with this revision. One comment, although not necessary to include in the manuscript, is that TMS will not always impact neural activity in a focal manner, as the effects of stimulation propagate throughout neural networks. The authors should consider this in potential future work (as described in the discussion).

Reviewer #4: (No Response)

Reviewer #5: (No Response)

7. PLOS authors have the option to publish the peer review history of their article (what does this mean?). If published, this will include your full peer review and any attached files.

Reviewer #1: No

Reviewer #3: No

Reviewer #4: No

Reviewer #5: No

---

## [Author Response · Author response to Decision Letter 1]

17 Nov 2020

Dear Dr. Javadi,

We thank you and the five reviewers for the evaluation of our revised manuscript entitled “Failure to modulate reward prediction errors in declarative learning with theta (6 Hz) frequency transcranial alternating current stimulation”. We have carefully reviewed the comments and have revised the manuscript accordingly. Below you can find our answers to the reviewers’ comments to which we responded in a point-by-point fashion. 

Sincerely,

Kate Ergo, Esther De Loof, Gillian Debra, Bernhard Pastötter, and Tom Verguts

Reviewer 1

Comment: Additionally, I appreciate that the distributional form of their data has been included in their figures, though I would recommend creating a clearer visual distinction in the data points in Fig. 2 panels (C) and (D), to separate incorrect vs. correct recognition (I believe they are current using triangle/circle markers, which are difficult to visually distinguish – the authors could consider jittering the vertical alignment to create more distinct columns of data points).

Response: We agree with Reviewer 1 that the triangle and circle markers were difficult to visually distinguish. Therefore, in the revised version of the manuscript, we have split up the Certainty panels of Figure 2 into four subfigures: certainty for correct recognitions in the real stimulation group, certainty for correct recognitions in the sham stimulation group, certainty for incorrect recognitions in the real stimulation group, and certainty for incorrect recognitions in the sham stimulation group. Each of these subfigures also contains individual data points representing mean certainty (gray circles for correct recognitions and gray triangles for incorrect recognitions) per condition for each participant. 

Comment: As I mentioned in my original review, the authors certainly have the flexibility to present additional data that can clarify or support their results. For example, within-subject behavioral responses could be presented that demonstrate the extent to which certain subjects may or may not have used the full dynamic range of certainty ratings.

Response: We added the within-subject behavioral responses for the certainty ratings as Supporting Information (S3 Certainty Ratings for subjects 1 to 20, S4 Certainty Ratings for subjects 21 to 41, S5 Certainty Ratings for subjects 42 to 61, S6 Certainty Ratings for subjects 62 to 77). 

Comment: Additionally, the authors could present a visualization that essentially captures the purpose of their LME, by showing the per-subject relationship between SRPE and certainty/accuracy. However, I would leave the need for such changes at the discretion of the editor.

Response: Because per-subject data can be noisy, we decided not to visualize the per-subject relationship between SRPE and Certainty/Accuracy. 

Reviewer 3

Comment: The authors have addressed all my concerns with this revision. One comment, although not necessary to include in the manuscript, is that TMS will not always impact neural activity in a focal manner, as the effects of stimulation propagate throughout neural networks. The authors should consider this in potential future work (as described in the discussion).

Response: We agree with Reviewer 3. We now added to the Discussion section on page 20 that: “Even though the spatial resolution of TMS remains debated (Slotnick, 2013), it is more focal than tACS.”

References

Slotnick, S. (2013). Controversies in cognitive neuroscience. New York: Palgrave Macmillan.

---

## [Editor Report · Decision Letter 2]

19 Nov 2020

Failure to modulate reward prediction errors in declarative learning with theta (6 Hz) frequency transcranial alternating current stimulation.

PONE-D-20-23792R2

Dear Dr. Ergo,

We’re pleased to inform you that your manuscript has been judged scientifically suitable for publication and will be formally accepted for publication once it meets all outstanding technical requirements.

Kind regards,

Amir-Homayoun Javadi, PhD

Academic Editor

PLOS ONE
---

## [Editor Report · Acceptance letter]

24 Nov 2020

PONE-D-20-23792R2 

Failure to modulate reward prediction errors in declarative learning with theta (6 Hz) frequency transcranial alternating current stimulation 

Dear Dr. Ergo:

I'm pleased to inform you that your manuscript has been deemed suitable for publication in PLOS ONE. Congratulations! Your manuscript is now with our production department. 

Kind regards, 

on behalf of

Dr. Amir-Homayoun Javadi 

Academic Editor

PLOS ONE